# *Pinus* Species as Prospective Reserves of Bioactive Compounds with Potential Use in Functional Food—Current State of Knowledge

**DOI:** 10.3390/plants10071306

**Published:** 2021-06-28

**Authors:** Marcin Dziedziński, Joanna Kobus-Cisowska, Barbara Stachowiak

**Affiliations:** 1Department of Gastronomy Sciences and Functional Foods, Poznan University of Life Sciences, Wojska Polskiego 28, 60-624 Poznan, Poland; 2Department of Technology of Plant Origin Food, Poznan University of Life Sciences, Wojska Polskiego 28, 60-624 Poznan, Poland; joanna.kobus-cisowska@up.poznan.pl (J.K.-C.); barbara.stachowiak@up.poznan.pl (B.S.)

**Keywords:** *Pinus*, pine, antioxidants, functional food, bioactive compounds

## Abstract

The pine (*Pinus* L.) is the largest and most heteromorphic plant genus of the pine family (*Pinaceae* Lindl.), which grows almost exclusively in the northern hemisphere. The demand for plant-based remedies, supplements and functional food is growing worldwide. Although pine-based products are widely available in many parts of the world, they are almost absent as food ingredients. The literature shows the beneficial effects of pine preparations on human health. Despite the wide geographical distribution of pine trees in the natural environment, there are very few data in the literature on the widespread use of pine in food technology. This study aims to present, characterise and evaluate the content of phytochemicals in pine trees, including shoots, bark and conifer needles, as well as to summarise the available data on their health-promoting and functional properties, and the potential of their use in food and the pharmaceutical industry to support health. Various species of pine tree contain different compositions of bioactive compounds. Regardless of the solvent, method, pine species and plant part used, all pine extracts contain a high number of polyphenols. Pine tree extracts exhibit several described biological activities that may be beneficial to human health. The available examples of the application of pine elements in food are promising. The reuse of residual pine elements is still limited compared to its potential. In this case, it is necessary to conduct more research to find and develop new products and applications of pine residues and by-products.

## 1. Introduction

*Pinus* (*Pinaceae*) is considered the largest genus of conifers, which includes more than 100 different species (Table 1 and Table 2) [1].

*Pinus* is a term first applied by Lineus in his work “Species Plantarum” for a group of 10 species, only five of which are currently included in this genus, i.e., *P. cembra, P. pinea*. *P. strobus, P. taeda* and *P. sylvestris* [3]. Because of the prevalence and morphological diversity of pines that can be found in many countries, many conflicting affiliations are known, particularly because many early affiliations to this genus were based on a very small number of morphological discriminants [3]. *Pinus* belongs to *Pinaceae* as a result of having shoot dimorphism, which includes short shoots (fascicles) that have one to eight narrow needles surrounded by bud scales at the base. Strong woody cone scales with the apical structure exposed after the first growing season (bump) and in the mature cone are also typical of the genus *Pinus*. Currently, *Pinus* is treated as a monophyletic taxon [1]. The subgenus *Pinus* (diploxylon or hard pines) has two fibrovascular bundles per needle, diverging pulvini at cataphyll bases (“fascicle breaks”), which usually have persistent sheaths. There are two to eight needles per fascicle and the position of the resin canals is polymorphic (septa; internal, medial external); the seed wings are articulated or oppressed [4]. In this subgenus, section *Trifoliae*, which is characterised by persistent fascicle sheaths, can be distinguished. Most species have cones with thick, woody scales that open at maturity; however, a few species have serotine pine cones. The section includes all North American hard pines, excluding *P. tropicalis* and *P. resinosa* [1]. The *Pinus* section has persistent fascicle sheaths. The number of needles ranges from one to three. External or medial resin canals are usually found [1]. Mature cones open at maturity (excluding *P. pinea*) and have thick scales. In most species, the seed wings are articulated; however, in *P. canariensis* and *P. roxburghii*, they have a decorative function. The section is widespread throughout Eurasia and the Mediterranean basin, as well as includes two species from the Americas: *P. resinosa* from eastern North America and *P. tropicalis* from western Cuba [1].

## 2. Nutritional Value and Mineral Content

Table 3 shows data on the nutritional value of different parts of trees of the genus *Pinus*. The nutritional value was identified in seeds, needles, bark and shoots.

The seeds have the highest energy value due to a high fat content [6]. The seeds also generally have the highest content of the tested nutrients, excluding vitamin C, which is higher in the conifer needles. The seeds of *P. pinea* can be a good source of Mg, P and especially Zn [6]. These seeds have higher zinc content than sesame seeds (approx. 4.5 mg/100 g) and seeds of some pumpkin species (0.54–1.31 mg/100 g), which are considered to be good dietary sources of zinc [13,14]. It is well known that different parts of plants have different nutritional content [15]. Seeds are generally lower in vitamins than the green parts of plants; however, they are higher in macronutrients, especially fats [16]. The uptake of mineral nutrients and their content in a plant depends not only on their content in the soil in the form available for plants, but also on the mutual quantitative ratio of individual mineral nutrients in the environment and on the afforestation level [17,18,19,20]. Other factors, such as soil pH, temperature, water supply, rainfall, access to sunlight, precipitation, weather and climate change, are also of great importance [21,22,23]. Nutrients, which can be categorized as macro- and micronutrients, have a nutritional role in plants [24]. Macronutrients affect biochemical processes, physiological responses and yield quantity [17,25]. When it comes to macronutrients, their role in plant organisms includes many life processes that determine plant functioning [24,26]. Therefore, it is very difficult to clearly indicate a specific role of elements because they act in a complex way. The role of micronutrients, on the other hand, is more specific, as it is related to specific, well-defined life processes in the plant and to plant growth [27,28]. Nutrient deficiency results in various disorders in terms of the normal growth and development of the plant [29,30]. Some nutrients, because of their specific functions in the plant, may limit the growth of certain pathogens [31]. Those constituents include zinc, sulphur, calcium and potassium [32]. Plant raw materials are a good source of minerals in the diet. This includes brews such as tea brews, coffee brews and herbal mixtures. As indicated by the results of many works, pine shoots can also be a valuable raw material for the preparation of brews in nutrition [33,34]. Pine seeds were found to be a good source of magnesium—an electrolyte essential for many metabolic and biological processes in the body, including acting as a cofactor in over 300 enzyme reactions [35]. Pine seeds were also found to be high in phosphorus and zinc, which are key minerals in terms of metabolic processes and energy metabolism [36]. Both the outer and inner bark is rich in resinous acids. These compounds may be toxic and allergenic; however, a positive effect has also been shown—abietic acid, which is found mainly in the inner bark, can act as an inhibitor of testosterone 5α-reductase [37]. Testosterone reductase inhibitors are used for treatment of benign prostatic hyperplasia, prostate cancer and pattern hair loss [38].

## 3. Polyphenol Content

Polyphenols are chemical compounds found in herbs, vegetables and fruit that have a wide range of uses. Currently, more than 8000 phenolic compounds are known. They include flavonoids, tannins, phenolic acids and their derivatives such as polymers [39]. Polyphenols are essential secondary metabolites that allow plants to grow and develop. They also protect plants from insects and other factors [38,40,41]. Polyphenols found in plants are involved in functions related to sensory properties such as colour, bitterness and sourness [42,43]. The presence of benzene rings and hydroxyl groups is common to all polyphenols. However, they are very diverse and can be divided into several subgroups. There are different ways to categorise these compounds based on their source of origin, biological function or chemical structure [39]. Polyphenols can be divided into different categories. Classifications are frequently used according to the number of present phenolic rings and structural components, which combine these rings, by differentiating the molecules into phenolic acids, flavonoids, stilbenes and lignans [44,45]. Simple phenols and flavonoids correspond to most natural phenolic substances. Moreover, flavonoids belong to the most common group of these compounds. Their common order is C6–C3–C6, which corresponds to two aromatic rings (rings A and B) bonded to three carbon atoms to produce an oxidised heterocycle (ring C). As a result of the type of hydroxylation and differences in the chromate ring (C ring), flavonoids can be further divided into distinct subgroups, including anthocyanins, flavan-3-ols, flavones, flavanones and flavonols [46,47,48]. The demand for phenolic acids is very high in many industries because they are used as precursors to other important bioactive molecules that are regularly needed for therapeutic and cosmetic purposes, as well as for food industry. Phenolic acids are also commercially available as dietary supplements [49].

Various parts of a pine (needles, seeds, bark and cones) and different solvents can be used to extract polyphenols. The pine bark is the best-examined part. Although all pine extracts have significant amounts of polyphenols, their content in the extract depends on the solvent type, extraction method, plant part used or pine species (Table 4). This results from natural variability, such as genotype, crop differences and harvesting conditions, climate, soil type, etc. [49,50]. Polyphenols were found to reduce morbidity and slow the progression of cardiovascular, neurodegenerative and cancer diseases. The mechanism of action of polyphenols is strongly associated with their antioxidant activity and reduction of reactive oxygen species in the human body [51,52]. Furthermore, the health-promoting properties of plant polyphenols include anti-inflammatory, anti-allergic, anti-atherosclerotic, anticoagulant and antimutagenic effects [53]. There are now pine tree preparations on the market, which are concentrated sources of polyphenols. The most popular pine tree preparation is an extract from *P. pinaster—*Pycnogenol^®^ (Horphag Research Ltd., Geneva, Switzerland). The quality of this extract is defined in the United States Pharmacopeia (USP 28). Between 65% and 75% of Pycnogenol are procyanidins comprising catechin and epicatechin subunits with varying chain lengths. Other constituents include polyphenolic monomers, phenolic or cinnamic acids and their glycosides. According to many studies, the constituents of Pycnogenol are highly bioavailable [54]. The daily intake of polyphenols among the general population ranges from 0.1 to 1.0 g per day. Fruit, vegetables, herbs, spices, coffee, tea and wine are the main source of polyphenols [55,56].

## 4. Essential Oils

Essential oils are volatile, natural, complex compounds with strong odours, which are generated by aromatic plants as secondary metabolites. They are usually obtained through steam or water distillation. Because of their known antiseptic, bactericidal, virucidal, fungicidal, medicinal and aromatic properties, they are used in the food industry and pharmacy to increase shelf life. They are also used as antibacterial, analgesic, sedative, anti-inflammatory, spasmolytic substances and local anaesthetics [65,66,67]. Most constituents of essential oils can be classified as lipophilic terpenoids, phenylpropanoids, or short-chain aliphatic hydrocarbon derivatives of low molecular weight; the former are the most common and characteristic. These include allylic, mono, bi- or tricyclic mono- and sesquiterpenoids from different classes that constitute the major part of essential oils, such as hydrocarbons, ketones, alcohols, oxides, aldehydes, phenols or esters [68,69]. Moreover, organic acids, phenols, coumarins, nitrogen and sulphuric substances are also found in essential oils. A single essential oil can have from 20 to 200 components, of which only one is ever dominant and gives a scent to the whole mixture of compounds. Variations in the composition of essential oils depend on environmental factors, plant varieties and the plant parts from which the oil is extracted [70]. Therefore, the chemical composition of oil is closely related to its storage conditions, as well as the environment in which the starting material was stored before its distillation. Since terpenes, i.e., α-pinene, are volatile and thermolabile, they are easily oxidised and hydrolysed [71,72,73]. The essential oil content is only a small percentage of the total weight of the plant. The oils can be found in plant cell tissue, glands or canals located in several parts of the plant (leaves, bark, roots, flowers, fruit, seeds). The presence of this mixture in living tissue is not fully explained. It is believed to be related to attracting insects that pollinate the plant or to repelling potential pests [74,75,76]. Pine essential oils are most frequently used as perfume and repellent ingredients. Turpentine is used for manufacturing many cosmetics, air fresheners and aromatherapy products [77,78,79]. The product that remains after distillation is the rosin that is a non-volatile fraction of oleoresins. It usually contains approx. 90% of resin acids and 10% of neutral components, monocarboxylic acids and diterpenoid acids, whereas the most common acids are abietic or pimaranic ones [80]. Rosin is used for manufacturing of adhesives, printer’s inks, soldering fluxes, varnishes and sealing waxes. It is also used as a glazing agent in many food products, including medicines and chewing gums [81].

Pine essential oils contain more than 50 ingredients. Their concentrations vary depending on the plant variety, crop, distillation method and part of the plant (Table 5). The following compounds are found in the greatest quantity: α-pinene, β-pinene, β-phellandrene, β-caryophyllene, camphene, α-terpineol, germacrene D, bornyl acetate, citronellol, β-caryophyllene and tricyclene [82,83,84]. Alpha-pinene (α-pinene) and beta-pinene (β-pinene) belong to bicyclic monoterpenes, commonly found in various species of pine trees of the genus *Pinus* [85]. Studies have shown that these phytochemicals exhibit diverse biological activity, which contributes to their various uses and applications. They can be used as fungicides, flavours and fragrances, as well as antiviral and antimicrobial agents [86]. The uses of α- and β-pinene go beyond therapeutic and nutritional applications. They are versatile compounds that are used in polymer synthesis [87]. Pinenes are generally recognised as safe (GRAS); thus, they are recognised by the U.S. Food and Drug Administration (FDA) as compounds that can be used in food products [88].

## 5. Antioxidant Activity

Free radicals and other reactive oxygen species, such as superoxides, hydroxyl radicals and hydrogen peroxide, are generated by either exogenous substances or endogenous metabolic processes of the human body, or in food products, react very rapidly with DNA, lipids and proteins, causing cell damage. Antioxidants, whose action is based on their ability to donate hydrogen atoms to free radicals, are compounds that protect against them [89,90]. In recent years, the interest in natural antioxidants has increased, which resulted in an intensification of research on them in various scientific fields. As a result, numerous articles concerning natural antioxidants, including polyphenols, flavonoids, vitamins and volatile compounds, have been published. Various assays were developed to evaluate the antioxidant activity of plants and food ingredients [91,92,93,94]. The use of at least two different methods of testing the antioxidant activity of samples is a generally accepted good practice. A combination of electron- or free radical capture assays, such as DPPH, ABTS, ACA or FRAP, as well as lipid peroxidation assays, is also recommended [95,96,97,98].

Different parts of trees (bark, needles, shoots, seeds), as well as various extraction methods and solvents, were used in the study on antioxidant properties of trees from the *Pinus* genus. As a result, extracts were correlated with various components and, thus, different antioxidant potentials, measured using multiple methods (Table 6). Several correlations may be observed. In the study of total polyphenol content using the Folin-Ciocalteu reagent, alcoholic extracts obtained from the tree bark, particularly from *P. radiata* (1610 mg of gallic acid equivalents/200 mL) and *P. brutia* (412.42 ± 7.56 mg gallic acid/g extract), adopting higher values compared to the aqueous extracts obtained from the shoots of *P. sylvestris* (0.86 ± 0.09 mg gallic acid/g dw) [7,99,100]. In free radical tests, which determined the value of EC50, the ethanolic extracts from *P. koraiensis* seeds (0.023 ± 0.004 mg/mL) and methanolic extracts from *P. bruti* bark (9.17 ± 0.13 μg/mL) assumed the lowest values of EC50, and thus, exhibited the strongest antioxidant activity [100,101]. Many existing studies explicitly state that the application of aqueous mixtures of water and organic solvents, such as ethanol, methanol, acetone, isopropanol or acetonitrile, significantly increases the antioxidant efficacy of many plant products [102]. In research studying the effect of solvent on the antioxidant activity of *P. densiflora* needle extracts using various concentrations of water and ethanol (0–100%), it was observed that 40% ethanolic needle extracts exhibited the highest radical scavenging capacity, followed by extracts containing 60%, 20%, 80%, 0% and 100% of ethanol, respectively [103]. Similar results were observed in the study on *P. densiflora* bark, which compared the content of phenolic compounds and antioxidant potential of extracts containing ethanol in the range of 0, 20, 40, 60, 80 or 100%, 20 or 40% of methanol, isopropanol and acetonitrile, as well as used acetone with distilled water (*v*/*v*) as extraction solvents [102]. Experiments revealed that bark extracts containing 20% of ethanol, 40% of ethanol and 20% of acetronile displayed the highest antioxidant potential and the highest content of phenolic compounds [102].

## 6. Pharmacological Properties

People around the world use herbal supplements and medicines due to their beneficial effects on human health [106]. Bark, needles, pollen and other parts of numerous pine species have been used for many years and proven to constitute excellent raw materials in the production of goods [107]. The first documented use of pine bark extracts dates back to 1535, when French explorer Jacques Cartier described events in which he and his crew avoided death from scurvy—a disease caused by vitamin C deficiency—by drinking pine bark brew. In 1951, French researcher Jacques Masquelier began studying herbal raw materials to identify their bioactive components. He was able to extract proanthocyanidins from the *P. pinaster* bark in the amount that could be used for manufacturing purposes [54]. Despite their acknowledged medicinal properties, the timber industry had regarded tree bark and shoots as inconvenient waste products; only in recent years have they been widely recognised as a rich source of natural polyphenols, containing potentially beneficial nutritional, health and medicinal properties [99]. Many standardised extracts of various pine species are currently used as dietary supplements and phytochemicals aiding in the treatment of various diseases around the world, including chronic inflammation, circulatory disorders and asthma (Table 7).

Several in vitro, animal and human studies have indicated the prophylactic and therapeutic effects of extracts from various pine species [107,108,109,110,111,112,113,114,115,116,117,118,119,120,121,122,123,124,125,126,127,128,129,130]. In a systematic review, published by the Cochrane Collaboration, which included 27 RCTs evaluating the effects of supplements containing pine bark extracts on 10 different chronic diseases: asthma, Attention Deficit Hyperactivity Disorder, cardiovascular disease and risk factors, chronic venous insufficiency, diabetes, erectile dysfunction, female sexual dysfunction, osteoarthritis, osteopenia and traumatic brain injury, it was concluded that small sample sizes, a limited number of RCTs, variability in outcome measures and poor reporting of the RCTs included, rendered it impossible to draw definitive conclusions about the efficacy or safety of supplements containing pine bark extract [108]. However, the aforementioned review did not take into account many other studies, including particularly interesting research on skin health and protection. Both the study on photoprotective and anti-photoaging effects presented a positive influence of *P. pinaster* bark extracts [109,110]. Furthermore, the role of antioxidants from the pine extracts in neuroprotective activity may prove to be fundamental, as *P. radiata* bark extracts exhibited effectiveness in two cases of RCT [111,112]. Nervous system inflammation and oxidative stress are believed to be the most characteristic symptoms of Alzheimer’s disease and play a key role in neurotoxicity. Thus, a suitable antioxidant strategy may improve the treatment of neurodegenerative diseases and dementia. Numerous studies have confirmed the neuroprotective effects of polyphenolic compounds, which protect neurons from the neurotoxin-induced injuries, as well as provide the ability to inhibit nervous system inflammations and the potential to advocate memory, learning and cognitive functions [113].

**Table 7 plants-10-01306-t007:** Pharmacological properties of *Pinus*.

Activity	Material	Experimental Model	Result	Source
Antihypertensive	*P. densiflora Sieb. et Zucc.* extract	A group of Wistar-Kyoto rats—a normotensive group—was orally administered tap water. Four groups of spontaneously hypertensive rats were orally administered tap water, captopril (a positive control), 50 mg/kg/day of KRPBE *P. densiflora* bark extract (Korean red pine bark extract; KRPBE) and 150 mg/kg/day of KRPBE, respectively. The blood pressure of rats was measured once a week during the seven weeks of oral administration of drugs. After seven weeks, the researchers collected the rats’ lungs, kidneys and serum, and subsequently determined the activity of angiotensin-converting enzyme (ACE), as well as the content of angiotensin II and malondialdehyde (MDA).	Blood pressure of rats served with captopril and KRPBE was significantly lower than that of the SHR control group. The activity of ACE, as well as the content of angiotensin II and MDA, was significantly lower in groups administered with captopril and KRPBE than those in the SHR control group.	[114]
Anti-adipogenic	*P. densiflora* aqueous bark extract	Four-week-old male C57BL/6 mice were fed with regular feed (18% kcal from fat) or HFD (60% kcal from fat). Animals fed with HFD were additionally subjected to PineXol treatment at 10 or 50 mg/kg body weight (PX10 or PX50, respectively).	Compared to the HFD group, the PX50 group was characterised by statistically lower body weight and body fat mass (*p* < 0.05 and *p* < 0.001, respectively). In the PX50 group, concentrations of hepatic triglycerides, total cholesterol and low-density lipoprotein cholesterol were lower than those in the HFD group (*p* < 0.01). The levels of acetyl CoA carboxylase (*p* < 0.01), elongase of a very long chain of fatty acids 6 (*p* < 0.01), stearoyl CoA desaturase 1 (*p* < 0.05), microsomal triglyceride transfer protein (*p* < 0.01) and sterol regulatory element-binding protein 1 (*p* < 0.05) in the PX50 group were significantly lower compared to their respective levels in the HFD group. In the white adipose tissue, the levels of CCAAT enhancer-binding protein alpha (*p* < 0.05), peroxisome proliferator-activated receptor gamma (*p* < 0.001) and perilipin (*p* < 0.01) in the PX50 group were lower than those in the HFD group.	[115]
Antidiabetic	*P. roxburghii* ethanolic bark extract	Rats were induced with diabetes through alloxan injection (120 mg/kg body weight). Control rats were either healthy and untreated, or induced with diabetes, untreated and provided only with distilled water. The acute effect of ethanolic extract was evaluated by administering 100, 300 and 500 mg/kg body weight p.o. of the extract to normoglycemic rats. In the chronic model, the ethanolic extract was administered to normal and alloxan-induced, diabetic rats at 100, 300 and 500 mg/kg body weight p.o. per day for 21 days. Levels of blood glucose and the values of body weight were monitored at specific intervals using different biochemical parameters.	Statistical data indicated a significant (*p* < 0.01) increase in the body weight, as well as a decrease in the level of blood glucose, glycated haemoglobin, total cholesterol and serum triglycerides. The level of HDL cholesterol was significantly (*p* < 0.01) increased in rats administered with the extract.	[116]
Hepatoprotective	*P. roxburghii* wood oil	The administration of *P*. *roxburghii* wood oil at 200, 300 and 400 mg/kg body weight was examined in terms of its hepatoprotective activity on rat liver damage induced by carbon tetrachloride and ethanol.	Noticeably high levels of serum aspartate aminotransferase, alanine aminotransferase, alkaline phosphatase, total bilirubin, malondialdehyde (MDA) and low levels of reduced glutathione (GSH) and total protein induced by hepatotoxins, significantly inclined towards adopting normal levels due to the wood oil administered at 200 and 300 mg/kg.	[117]
Antidyslipidemic	*P. roxburghii* needles, hexane (B), chloroform (C), n-butanol soluble (D) and n-butanol insoluble (E) fractions.	Dyslipidemic hamsters were divided into six groups and fed with five solvent fractions (A, B, C, D and E) of *P*. *roxburghii* needles.	Extract from *P*. *roxburghii* needles exhibited the significant potential to decrease the level of the plasma lipid profile, as well as having a beneficial effect on the HDL-C and its ratio with total cholesterol in a dyslipidemic hamster model.	[118]
Analgesic	*P. roxburghii* Sarg. stem bark ethanolic extract	Analgesic activity was evaluated using acetic acid-induced writhing and tail immersion tests in Swiss albino mice.	Alcoholic extract from *Pinus roxburghii* Sarg. (at 100, 300 and 500 mg/kg) significantly and dependently reduced the number of abdominal constrictions induced in mice by administering a 1% solution of acetic acid. This dose-dependent protective effect reached a maximum pain inhibition of 80.95% at 500 mg/kg.	[119]
Anticonvulsant	*P. roxburghii* alcoholic extract	Anticonvulsant activity was evaluated by means of maximal electroshock (MES) and pentylenetetrazole-induced (PTZ) seizures in Wistar albino rats at various doses (i.e., 100, 300 and 500 mg/kg).	In the MES-induced seizure model, AEPR at 300 and 500 mg/kg body weight significantly reduced all phases of convulsion (*p* < 0.01). In the PTZ-induced seizure model, administration of the extract at 300 and 500 mg/kg half an hour before the injection of PTZ significantly delayed the onset of clonic seizures (*p* < 0.01).	[120]
Anti-viral (HIV-1)	*P. pinaster* ssp. atlantica extract (Pycnogenol)	The inhibitory effect of the extract on virus binding to MT-4 cells was examined by infecting the MT-4 cells with IIIB-env-Hiv-1 in the presence or absence of extract.	Addition of the compound at the time of injection resulted in a dose-dependent inhibition of the cytopathic effect, as well as a dose-dependent reduction in p24.	[121]
Anti-viral (Epstein-Barr virus)	*P. massoniana* aqueous bark extract	Inhibition of the immediate-early viral gene transpiration by the extract was assessed by transient transfection assay.	*P. massoniana* bark extract (PMBE) at 60 microg/mL or a higher dose, inhibits the expression of the Epstein-Barr virus (EBV) lytic proteins, such as Rta, Zta and EA-D. The EBV lytic cycle was blocked by the inhibition of the immediate-early gene transcription.	[122]
Wound healing	Methanol and *P. longifolia roxburghii* aqueous leave extracts	Extracts were examined in terms of wound healing properties on excision and incision wound models in Wistar albino rats.	Both extracts exhibited significant wound healing activity. However, the rate of wound contraction and epithelialisation was faster in groups administered with methanol extract.	[123]
Anti-cancer	*P. roxburghii* essential oil	The essential oil was tested against human cancer cell lines, i.e., cultured HCT-116 (colon cancer), KBM-5 (myelogenous leukaemia), U-266 (multiple myeloma cells), MiaPaCa-2 (pancreatic cancer cells), A-549 (lung carcinoma cells) and SCC-4 (squamous cell carcinoma) cell lines by means of the MTT assay.	The percentage inhibition of PREO activity was found to be concentration-dependent. U-266 exhibited maximum inhibition of 83%, while HCT-116, SCC4, MiaPaCa-2, A-549 and KBM-5 manifested 71, 69, 73, 73 and 76% of inhibition, respectively.	[124]
	Petroleum ether, ethyl acetate, chloroform and *P. roxburghii* Sarg. ethanolic extract	Effect of *Pinus roxburghii* Sarg. extracts on the growth of human IMR32 neuroblastoma cancer cell line was studied using the SRB assay.	Petroleum ether and chloroform extracts were the only extracts that exhibited anticancer activity.	[125]
Cardio-protective	*P. pinaster* ssp. atlantica extract (Pycnogenol)	Twenty-three patients with coronary artery disease (CAD) completed this randomised, double-blind, placebo-controlled cross-over study. Apart from the standard cardiovascular therapy, patients received Pycnogenol (200 mg/day) for 8 weeks followed by the placebo, or vice versa. At a baseline and after each treatment period, the endothelial function, assessed in a non-invasive manner via flow-mediated dilatation (FMD) of the brachial artery using high-resolution ultrasound, biomarkers of oxidative stress and inflammation, platelet adhesion and 24 h blood pressure monitoring were evaluated.	In CAD patients, treatment with Pycnogenol was associated with an improvement of FMD from 5.3 ± 2.6 to 7.0 ± 3.1 (*p* < 0.0001), while no change was observed in case of placebo (5.4 ± 2.4 to 4.7 ± 2.0; *p* = 0.051). Isoprostane—which influences the oxidative stress index—significantly decreased from 0.71 ± 0.09 to 0.66 ± 0.13 after treatment with Pycnogenol, while no change was observed in the group treated with placebo (mean difference 0.06 pg/mL with an associated 95% CI (0.01, 0.11), *p* = 0.012). Inflammation markers, platelet adhesion and blood pressure levels did not change following the treatment with Pycnogenol or placebo.	[126]
Neuroprotective	*P. densiflora* aqueous bark extract	Neuroprotective effect (anticholinesterase activity) was determined using the AChE and BChE assays while intracellular oxidative stress was evaluated using the fluorescent assay using DCFH-DA on neuronal PC-12 cells.	Pretreatment of PC-12 cells with Kextract decreased the oxidative stress in a dose-dependent manner compared to cells exposed solely to oxidative stress. Inhibition of AChE and BChE occurred at 10 µg/mL and 100 µg/mL in TE values—approx. 68.3 nM and 15.1 nM for the inhibition of AChE and BChE, respectively.	[127]
	*P. roxburghii* Sarg. methanolic extract	The in vitro cell viability activity of *P. roxburghii* was assessed using the PC-12 cell lines. The in vivo neuroprotective activity of *P. roxburghii* was tested on Wistar albino rats (both sexes). ICV-STZ (3 mg/kg, bilateral) was administered to induce a memory deficit.	*P*. *roxburghii* exhibited significant cell viability at 10, 50 and 100 µg/mL in an in vitro assay on PC-12 cell lines. In the in vivo activity, ICV-STZ significantly deteriorated memory, cognition, tissue oxidative stress and the AchE activity. *P*. *roxburghii* (at 100, 200 and 300 mg/kg p.o.) and donepezil (at 3 mg/kg, p.o.) significantly (*p* < 0.05) reversed the behavioural changes in rats when tested in a morris water maze and elevated plus maze. Increased levels of lipid peroxidation, AchE activity and decreased the level of glutathione were significantly (*p* < 0.05) antagonised by *P*. *roxburghii*, similarly to the case of donepezil in rat brain.	[128]
	*P. radiata* bark	Sixty adults who sustained a mild TBI 3–12 months before recruitment and were experiencing persistent cognitive difficulties (CFQ score > 38), were randomised in order to receive enzogenol (1000 mg/day) or a corresponding dose of placebo for 6 weeks. Subsequently, all participants received enzogenol for a further 6 weeks, followed by placebo for 4 weeks. Compliance, side-effects, cognitive failures, working and episodic memory, post-concussive symptoms and mood were evaluated at baseline, as well as in the 6th, 12th and 16th week.	Enzogenol was found to be safe and well-tolerated. Trend and breakpoint analyses revealed a significant reduction in cognitive failures after 6 weeks (mean CFQ score, 95% CI, Enzogenol versus placebo 6.9 (10.8 to 4.1)). Improvements in the frequency of self-reported cognitive failures were estimated to continue until the 11th week before stabilising.	[111]
	*P. radiata*	During the period of 5 weeks, the participants (42 males aged 50–65) were supplemented either with Enzogenol combined with vitamin C, or vitamin C only. A battery of computerised cognitive tests was administered while cardiovascular and haematological parameters were assessed before and after supplementation.	The speed of the response to the spatial working memory and immediate recognition tasks improved after supplementation with Enzogenol combined with vitamin C, whereas supplementation with vitamin C alone did not induce any improvement. A trend in the reduction of systolic blood pressure was observed in patients supplemented with Enzogenol combined with vitamin C, but not with vitamin C alone. The blood safety parameters remained unchanged.	[112]
Photoprotective	*P. pinaster*	A total of 21 volunteers were administered oral supplementation of Pycnogenol: 1.10 mg/kg body weight (b. wt.)/day (d) for the first 4 weeks and 1.66 mg/kg b. wt./d for the following 4 weeks. The minimal erythema dose (MED) was measured twice before the supplementation (baseline MED), once after the first 4 weeks of supplementation and the last time at the end of the study.	An increase in MED was observed after supplementation with 1.10 mg/kg b. wt./d of PBE for 4 weeks (mean MED 5 34.62 mJ/cm^2^, 95% CI 5 from 31.87 to 37.37). A supplementation with 1.66 mg/kg b. wt./d of PBE for the last 4 weeks of the study caused an even further increase in MED (mean MED 5 39.62 mJ/cm^2^, 95% CI 5, from 37.51 to 41.73).	[109]
Anti-photoaging	*P. pinaster*	A total of 112 women with mild to moderate skin photoaging symptoms were randomised to either take part in a 12-week open trial regimen of 100 mg PBE supplementation once a day or to be in a parallel-group—a trial regimen of 40 mg PBE supplementation once a day.	A significant decrease in clinical grading of skin photoaging scores was observed during both 100 mg and 40 mg of PBE daily supplementation regimens. Furthermore, a significant reduction in the pigmentation of age spots was demonstrated using skin colour measurements.	[110]

Abbreviations: KRPBE—Korean red pine bark extract; SHR—spontaneously hypertensive rats; ACE—angiotensin-converting enzyme; MDA—malondialdehyde; HFD—high fat diet, PX—PineXol; GSH—glutathione; AEPR—alcoholic extract of bark of *Pinus roxburghii* Sarg.; MES—maximal electroshock; PTZ—pentylenetetrazole; EBV—Epstein-Barr virus; PREO—*P*. *roxburghii* essential oil; MED—minimal erythema dose; CAD—coronary artery disease; FMD—flow-mediated dilatation; AChE—acetylcholinesterase; BChE—butyrylcholinesterase; ICV-STZ—Intracerebroventricular Streptozotocin Injections; CFQ—Cognitive Failures Questionnaire; b. wt./d—body weight/day; PBE—pine bark extract.

## 7. Antimicrobial Activity

The increasing incidence of infectious diseases, severe side effects related to the intake of many antibiotics and the development of antibiotic resistance substantiate the growing interest in the identification of new antimicrobial compounds, both natural and synthetic agents [131,132,133]. Plant resin has been applied to treat diseases in folk medicine for thousands of years. It was also used in the pharmaceutical industry before the introduction of modern antibiotics. Many of the secondary metabolites of trees adopt a protective function against predators and pathogenic microorganisms. The antimicrobial activity of extracts, oils and resins from trees of the *Pinus* genus may be related to various organic compounds, such as alkaloids, phenols and terpenes (Table 4 and Table 5) [82,134]. The discovery of biological effects of the *Pinus* spp. compounds suggests that they may be applied in the creation of environmentally friendly and biocompatible pharmaceuticals.

The most common human pathogen, colonising one-third of healthy people throughout the world, is *Staphylococcus aureus* [135]. *S. aureus* is also an etiologic agent contributing to the development of many human infections, including pneumonia, meningitis, toxic shock syndrome, bacteremia and endocarditis. *S. aureus* is further known for its rapidly advancing resistance to antibiotics [136,137,138,139]. The studies proved that extracts and essential oils from *P. cembra*, *P. koraiensis*, *P. brutia*, *P. densiflora* and *P. sylvestris* inhibit the growth of many *S. aureus* strains, including: ATCC 25923, 25923, 503, 29213, ATCC BAA-977 and ATCC 13565 (Table 8) [104,140,141,142]. Trees from the *Pinus* species display properties aiding in the fight against many strains of various bacteria. The highest inhibition was observed in *M. luteus* NRRL B-4375, *Proteus vulgaris* ATCC 13315, *Shigella flexneri* AT CC 12026 and *Streptococcus faecalis* ATCC 19433 [141,142]. *Shigella flexneri* is a gram-negative bacterium causing the most contagious bacterial shigellosis. Shigellosis generates 1.1 million deaths and more than 164 million cases each year. The majority of said cases involve children in developing countries. Pathogenesis of *S. flexneri* is based on its ability to invade and replicate within the colonic epithelium, leading to severe inflammation and destruction of the epithelium itself [143]. Despite intensive research, conducted for over 60 years using various vaccination strategies, a safe and effective vaccine is not yet available [144]. Numerous studies indicate that plant secondary metabolites can inhibit the spread of phytopathogens, by acting both as antimicrobial agents and elicitors of other defensive responses. Many of the aforementioned metabolites negatively affect the clinically relevant pathogens and their use as “antibiotic enhancers” or “virulence attenuators” fighting against infectious diseases in humans is promising [145].

Compounds extracted from the trees of the *Pinus* genus presented in many studies exhibited different levels of antimicrobial activity against yeast, gram-positive and gram-negative bacteria, which validates the traditional application of these substances [140]. Additionally, such extracts, oils and resins display the insecticidal, phytotoxic and antioxidant potential [141]. Therefore, it is necessary to conduct research aided by biological studies with recovery, identification and testing of a single compound and/or multiple compounds to determine its/their biological effects [142].

## 8. Food Application of *Pinus*

There is an increasing demand for health-promoting plant products all over the world [149]. Today, conifer shoots are virtually unused as a food ingredient, despite their common availability in many parts of the world. An exception is a common juniper, whose berry-like cones are a valued seasoning in Europe [150]. Pine shoot products, such as pine shoot syrup, pine shoot-based beer and herbal teas are available on the market. Despite its many potential applications, currently, the shoot products are not very popular [151].

To date, there has been little research on the use of pine tree elements in food products (Table 9). However, current literature indicates a possible application of such ingredients in beverages, dairy products, meat products or even bread. The addition of *P. pinaster* extracts increases the antioxidant potential of juices and dairy products. With regard to juices, polyphenols derived from pine extracts may also have a negative, inhibitory effect on the microflora [151,152,153,154]. Moreover, in terms of sensory experience, kefir enriched with pine bud syrup was assessed higher than the control sample, which indicates that it may also serve as an ingredient providing flavour and aroma [151]. In the case of the addition of pine extract to bread and meat, the substance acted as a shelf life extender by inhibiting the growth of bacteria and oxidisation of fats [155,156]. Moreover, pine extracts can be possibly applied in the future as additives and preservatives, as they are commercially sold as dietary supplements. Many of these extracts are listed on the Everything Added to Food in the United States (EAFUS) database that the Food and Drug Administration (FDA) approved as food additives or affirmed as Generally Recognised as Safe (GRAS) [157].

## 9. Conclusions

Residues and by-products constitute an important source of industrially significant biocomponents. Various species of pine tree contain different compositions of bioactive compounds. However, even though the pine bark extracts are commercially available, there is no universal method of extraction that is suitable for all phenols. Depending on the ultimate goal of extraction, an individual examination should be performed to ensure the most appropriate extraction procedure. Regardless of the solvent, method, pine species and plant part used, all pine extracts contain a high number of polyphenols. Nevertheless, individual compounds are characterised by different concentrations, types and levels of their bioactivity. There are few studies on the identification and even fewer studies presenting the quantitative determination of individual polyphenols contained in pine extracts. Pine tree extracts exhibit several described biological activities that may be beneficial to human health. The available examples of the application of pine elements in food are promising. Pine tree extracts, syrups and other intermediates may be components that impart functional properties, extend the shelf life and assign desirable qualities to food products. Pine extracts and oils exhibit great potential as formulation ingredients for food, cosmeceutical and pharmaceutical industries. The reuse of residual pine elements is still limited compared to its potential. In this case, it is necessary to conduct more research to find and develop new products and applications of pine residues and by-products.

## Figures and Tables

**Table 1 plants-10-01306-t001:** Taxonomic hierarchy of genus *Pinus* L. [2].

Kingdom	Plantae
Subkingdom	Viridiplantae
Infrakingdom	Streptophyta
Superdivision	Embryophyta
Division	Tracheophyta
Subdivision	Spermatophytina
Class	Pinopsida
Subclass	Pinidae
Order	Pinales
Family	*Pinaceae*
Genus	*Pinus* L.

**Table 2 plants-10-01306-t002:** Classification of subgenus *Pinus* [1].

Section *Pinus*		Section Trifoliae		
Subsection *Pinus*	Subsection Pinaster	Subsection Contortae	Subsection Australes	Subsection Ponderosae
*P. densata, densiflora, hwangshanensis, kesiya, luchuensis, massoniana, merkusii, mugo, nigra, resinosa, sylvestris, tabuliformis, taiwanensis, thunbergii, tropicalis, uncinata, yunnanensis*	*P. brutia, canariensis, halepensis, heldreichii, pinaster, pinea, roxburghii.*	*P. banksiana, clausa, contorta, virginiana;*	*P. attenuata, caribaea, cubensis, echinata, elliottii, glabra, greggii, herrerae, jaliscana, lawsonii, leiophylla, lumholtzii, muricata, occidentalis, oocarpa, palustris, patula, praetermissa, pringlei, pungens, radiata, rigida, serotina, taeda, tecunumanii, teocote*	*P. cooperi, coulteri, donnell-smithii, devoniana, douglasiana, durangensis, engelmannii, hartwegii, jeffreyi, maximinoi, montezumae, nubicola, ponderosa, pseudostrobus, sabineana, torreyana, washoensis.*

**Table 3 plants-10-01306-t003:** Nutritional value and mineral content.

Index	Species	Part of the Tree	Content	Reference
Energy value	*P. contorta* L.	needles	500 kcal/100 g	[5]
Energy value	*P. pinea* L.	seeds	583 kcal/100 g	[6]
Dry mass	*P. sylvestris* L.	shoots	13.98%	[7]
	*P. taeda* L.	stem	30.74%	[8]
		needles	1.55%	[8]
crude protein	*P. contorta* L.	needles	3.63%	[5]
crude protein	*P. pinea* L.	seeds	31.6 g/100 g	[6]
fat	*P. pinea* L.	seeds	44.9 g/100 g	[6]
triglycerides	*P. sylvestris* L.	inner bark	33.40 mg/g	[9]
		outer bark	1.71 mg/g	[9]
		conifer needles	10.3 µmol/g dry weight	[10]
Mono- and diglycerides of fatty acids		inner bark	2.26 mg/g	[9]
		outer bark	5.46 mg/g	[9]
		conifer needles	2.3 µmol/g dry weight	[10]
steryl esters		inner bark	1.54 mg/g	[9]
		outer bark	0.19 mg/g	[9]
free fatty acids		inner bark	0.63 mg/g	[9]
		outer bark	1.68 mg/g	[9]
		conifer needles	10.3 µmol/g	[10]
resin acids		inner bark	7.16 mg/g	[9]
		outer bark	2.39 mg/g	[9]
sterols and triterpenic alcohols		inner bark	4.50 mg/g	[9]
		outer bark	2.98 mg/g	[9]
fatty alcohols		inner bark	1.33 mg/g	[9]
		outer bark	1.25 mg/g	[9]
carbohydrates	*P. pinea* L.	seeds	13.3 g/100 g	[6]
total soluble sugar	*P. pinea* L.	seeds	5.15 g/100 g	[6]
reducing sugar	*P. pinea* L.	seeds	0.7 g/100 g	[6]
glucose	*P. sylvestris* L.	needles	121.8 µmol/g	[10]
fructose	*P. sylvestris* L.	needles	151.3 µmol/g	[10]
galactose/arabinose	*P. sylvestris* L.	needles	5.2 µmol/g	[10]
sucrose	*P. sylvestris* L.	needles	59.6 µmol/g	[10]
sucrose	*P. pinea* L.	seeds	4.3 g/100 g	[6]
raffinose/melibiose	*P. sylvestris* L.	needles	4.1 µmol/g	[10]
starch	*P. sylvestris* L.	needles	124.8 µmol/g	[10]
Na	*P. pinea* L.	seeds	11.7 g/100 g	[6]
Ca	*P. pinea* L.	seeds	13.8 mg/100 g	[6]
Ca	*P. sylvestris* L.	bark	0.38%	[11]
Ca	*P. sylvestris* L.	needles	0.53%	[12]
Ca	*P. taeda* L.	stem	0.09%	[8]
Ca	*P. taeda* L.	needles	0.31%	[8]
K	*P. pinea* L.	seeds	713 mg/100 g	[6]
K	*P. sylvestris* L.	Needles	0.54%	[12]
K	*P. sylvestris* L.	bark	0.172%	[11]
K	*P. taeda* L.	stem	0.08%	[8]
K	*P. taeda* L.	needles	0.54%	[8]
Mg	*P. pinea* L.	seeds	325 mg/100 g	[6]
Mg	*P. sylvestris* L.	Needles	0.09%	[12]
Mg	*P. sylvestris* L.	bark	0.059	[11]
Mg	*P. taeda* L.	stem	0.14%	[8]
Mg	*P. taeda* L.	needles	0.18%	[8]
P	*P. pinea* L.	seeds	512 mg/100 g	[6]
S	*P. sylvestris* L.	Needles	0.095%	[12]
Fe	*P. pinea* L.	seeds	10.2 mg/100 g	[6]
Fe	*P. sylvestris* L.	Needles	61.7 µg/g	[12]
Mn	*P. pinea* L.	seeds	6.9 mg/100 g	[6]
Mn	*P. sylvestris* L.	Needles	275.6 µg/g.	[12]
Zn	*P. pinea* L.	seeds	6.4 mg/100 g	[6]
Zn	*P. sylvestris* L.	Needles	53.63 µg/g	[12]
Cu	*P. pinea* L.	seeds	1.5 mg/100 g	[6]
Cu	*P. sylvestris* L.	Needles	5.3 µg/g	[12]
Cu	*P. sylvestris* L.	bark	2.98 mg/kg	[11]
N	*P. sylvestris* L.	bark	0.49%	[11]
N	*P. taeda* L.	stem	0.35%	[8]
N	*P. taeda* L.	needles	1.39%	[8]
ascorbic acid	*P. pinea* L.	seeds	2.5 mg/100 g	[6]
ascorbic acid	*P. sylvestris* L.	shoots	29.3 mg/g	[7]
Thiamine	*P. pinea* L.	seeds	1.5%	[6]
Riboflavin	*P. pinea* L.	seeds	0.28%	[6]

**Table 4 plants-10-01306-t004:** Polyphenol content.

Compound	Species	Part of the Tree	Content	Reference
gallic acid	*P. sylvestris* L.	shoots	208.38 ± 069 µg/g dw	[7]
2,5-dihydroxybenzoic acid			16.63 ± 0.54 µg/g dw	[7]
4-hydroxybenzoic acid			1084.92 ± 39.04 µg/g dw	[7]
caffeic acid			1502.03 ± 52.53 µg/g dw	[7]
syringic acid			145.44 ± 3.28 µg/g dw	[7]
p-coumaric acid			387.89 ± 15.83 µg/g dw	[7]
ferulic acid			2088.89 ± 56.89 µg/g dw	[7]
chlorogenic acid			518.25 ± 4.90 µg/g dw	[7]
sinapic acid			54.09 ± 2.06 µg/g dw	[7]
*t-*cinnamic acid			111.44 ± 3.4 µg/g dw	[7]
vanillic acid			0.46 ± 0.01 µg/g dw	[7]
salicylic acid			0.36 ± 0.00 µg/g dw	[7]
naringenin			1.59 ± 0.02 µg/g dw	[7]
vitexin			0.61 ± 0.01 µg/g dw	[7]
rutin			0.63 ± 0.02 µg/g dw	[7]
quercetin			0.98 ± 0.03 µg/g dw	[7]
apigenin			0.30 ± 0.01 µg/g dw	[7]
kaempferol			0.38 ± 0.01 µg/g dw	[7]
luteolin			0.30 ± 0.01 µg/g dw	[7]
protocatechuic acid	*P. radiata*	bark	46.2 ± 1.1 µg/mg	[57]
	*P. sibirica*	seeds	49.2 ± 0.5 mg/100 g dw	[58]
(+)-Catechin			52.5 ± 0.6 mg/100 g dw	[58]
vanillic acid			85.5 ± 1.0 mg/100 g dw	[58]
epigallocatechin gallate			47.0 ± 1.4 mg/100 g dw	[58]
syringic acid			101 ± 0.3 mg/100 g dw	[58]
()-epicatechin;			125 ± 3.1 mg/100 g dw	[58]
taxifolin			172 ± 3.1 mg/100 g dw	[58]
eriodictyol			383 ± 1.0 mg/100 g dw	[58]
(E)-cinnamic acid			12.2 ± 1.2 mg/100 g dw	[58]
naringenin			37.0 ± 2.1 mg/100 g dw	[58]
catechin	*P. sinaster*	bark	117.0 ± 8.0 mg/L	[59]
gallocatechin			16.8 ± 4.9 mg/L	[59]
taxifolin			447.7 ± 32.5 mg/L	[59]
quercetin			105.5 ± 2.7 mg/L	[59]
3,4 hydroxybenzoic acid			17.3 ± 2.4 mg/L	[59]
gallic acid			3.6 ± 0.7 mg/L	[59]
caffeic acid			20.6 ± 1.1 mg/L	[59]
o-coumaric acid			47.5 ± 25.3 mg/L	[59]
ferulic acid			47.2 ± 0.8 mg/L	[59]
rosmarinic acid			72.5 ± 4.0 mg/L	[59]
ellagic acid			402.2 ± 51.4 mg/L	[59]
naringin			173.4 ± 55.5 mg/L	[59]
apigenin			53.9 ± 0.1 mg/L	[59]
resveratrol			40.0 ± 0.4 mg/L	[59]
trans-ferulic acid	*P. radiata*	bark	5.9 ± 0.1 µg/mg	[57]
trans-caffeic acid			2.6 ± 0.1 µg/mg	[57]
()-epicatechin;			21.6 ± 1.7 µg/mg	[57]
(+)-Catechin			198.5 ± 6.4 µg/mg	[57]
cis-taxifolin			73.6 ± 2.7 µg/mg	[57]
trans-taxifolin			382.5 ± 12.1 µg/mg	[57]
quercetin			15.2 ± 1.0 µg/mg	[57]
quercetin, resin acid (abietic acid, neoabietic acid), taxifolin, catechin, quercetin derivative, taxifolin derivative, catechin and gallocatechin, kaempferol, rhamnetin isorhamnetin, myricetin, 3,4-dihydroxybenzoic acid, 3,4-dihydroxycinnamic acid, pinosylvin 3-methyl ether, dihydromonomethyl pinosylvin, resveratrol, glycoside, pinoresinol, secoisolariciresinol	*P. wallichiana and P. roxburghii, P. gerardiana*	stem and needle extract	presence found	[60,61]
1,5-diliydroxy-3,6,7-triniethoxy-8-allyloxyxanthone, 1-hydroxy-3,6-diinethoxy-2-β glucopyranoxanthone, friedelin, ceryl alcohol, b-sitosterol, taxifolin, quercetin, catechin, kaempferol, rhamnetin, 3,4-dihydroxybenzoic acid, 3,4-dihydroxycinnamic acid, pinosylvin, pinoresinol, resin acid, sterols, gallocatechin and tannins was found. hexacosyl ferulate	*P. roxburghii*	bark	presence found	[62,63]
12-hydroxydodecanoic acid, 14-hydroxytetradecanoic acid and 16-hydroxy-hexadecanoic acid		needle wax	presence found	[64]

Abbreviation dw—dry weight.

**Table 5 plants-10-01306-t005:** The composition of essential oils extracted from pine [82].

Part of the Plant	Bioactive Components	Average Concentration (%)
Needles	α-pinene	31.6
	β-pinene	13.8
	β-phellandrene	9.8
	germacrene D	9.2
	α-Terpineol	6.2
	camphene	7.7
	bornyl acetate	4.4
twigs	β-phellandrene	34.4
	α-pinene	17.7
	β-pinene	17.4
	germacrene D	6.5
	bornyl acetate	4.3
	camphene	3.2
	α-Terpineol	2.1
Needles and twigs	Tricyclene, Sabinene, Myrcene, 3-Carene, β-Z-Ocimene, γ-Terpinene, Terpinolene, E-Pinene hydrate, α-Campholenal, iso-3-Thujanol, Z-Verbenol, Borneol, Terpinene-4-ol, Myrtenal, E-Piperitol, Linalool acetate, α-Terpineol acetate, α-Copaene, β-Bourbonene, β-Elemene, β-Caryophyllene, β-Copaene, α-E-Bergamotene, α-Humulene, Z-Muurola-4(14),5-diene, γ-Cadinene, δ-Cadinene, α-Cadinene, E-Nerolidol, Germacrene-4-ol, Spathulenol, Caryophyllene oxide, Humulene epoxide II, Z-Cadin-4-en-7ol, Cubenol, α-Muurolol, α-Cadinol, Eudesma-4(15),7-diene-1-β-ol, Oplopanone, Cembrene	<1

**Table 6 plants-10-01306-t006:** Antioxidant properties of various *Pinus* species.

Method	Species	Material	Result	Reference
Total phenolic content	*P. koraiensis*	Seed 40% ethanolic extract	264 ± 10.52 mg of gallic acid equivalents/g	[101]
	*P. pinaster*	Bark ethanolic extract	890 mg of gallic acid equivalents/200 mL	[99]
	*P. radiata*	Bark ethanolic extract	1610 mg of gallic acid equivalents/200 mL	[99]
	*P. cembra* L.	Bark 80% aqueous methanol extract	299.3 ± 1.4 mg of gallic acid/g extract	[104]
	*P. cembra* L.	Needle 80% aqueous methanol extract	78.22 ± 0.44 mg of gallic acid/g extract	[104]
	*P. sylvestris* L.	Shoot aqueous extract	0.86 ± 0.09 mg of gallic acid/g dw	[7]
	*P. sylvestris* L.	Air-dried shoot 40% aqueous ethanol extract	13.4 ± 4.07 mg of gallic acid/g dw	[105]
	*P. sylvestris* L.	Vacuum-dried shoot 40% aqueous ethanol extract	8.34 ± 2.01 mg of gallic acid/g dw	[105]
	*P. sylvestris* L.	Freeze-dried shoot 40% aqueous ethanol extract	5.73 ± 2.55 mg of gallic acid/g dw	[105]
	*P. brutia*	Bark 80%aqueous methanol extract	412.42 ± 7.56 mg of gallic acid/g extract	[100]
OH scavenging activity EC50	*P. koraiensis*	Seed 40% ethanolic extract	0.391 ± 0.055 mg/mL	[101]
	*P. brutia*	Bark 80%aqueous methanol extract	0.5 ± 0.0 mg/mL	[100]
DPPH radical scavenging activity	*P. koraiensis*	Seed 40% aqueous ethanol extract	EC50 value 0.023 ± 0.004 mg/mL	[101]
	*P. cembra* L.	Bark 80% aqueous methanol extract	EC50 value 71.1 ± 0.5 μg/mL	[104]
	*P. cembra* L.	Needle 80% aqueous methanol extract	EC50 value 186.1 ± 1.7 μg/mL	[104]
	*P. sylvestris* L.	Shoot aqueous extract	200.94 ± 23.47 mg of gallic acid/g dw	[7]
	*P. sylvestris* L.	Air-dried shoot 40% aqueous ethanol extract	332.25 ± 10.49 dw μM Trolox/g dw	[105]
	*P. sylvestris* L.	Vacuum-dried shoot 40% aqueous ethanol extract	299.72 ± 15.97 dw μM Trolox/g dw	[105]
	*P. sylvestris* L.	Freeze-dried shoot 40% aqueous ethanol extract	339.00 ±19.61 dw μM Trolox/g dw	[105]
	*P. radiata*	Aqueous bark extract	36.3 ± 5.0% at 2.0 μg/mL	[57]
	*P. brutia*	Bark 80%aqueous methanol extract	1.47 ± 0.02 *Trolox equivalent* mg/mL	[100]
O2 inhibition activity	*P. sylvestris* L.	Vacuum-dried shoot 40% aqueous ethanol extract	8.34 ± 2.01 mg of gallic acid/g dw	[105]
ABTS radical cation scavenging assay	*P. sylvestris* L.	Freeze-dried shoot 40% aqueous ethanol extract	5.73 ± 2.55 mg of gallic acid/g dw	[105]
	*P. cembra* L.	Needle 80% aqueous methanol extract	0.3 ± 0.0 μM Trolox equivalent to 1 μg/mL extract	[104]
	*P. radiata*	Aqueous bark extract	55.1 ± 5.8% at 1.0 ug/mL	[57]
Reducing power assay EC50	*P. cembra* L.	Bark 80% aqueous methanol extract	26.0 ± 0.3 mg/mL	[104]
	*P. cembra* L.	Needle 80% aqueous methanol extract	104 ± 2 mg/mL	[104]
	*P. brutia*	Bark 80%aqueous methanol extract	9.17 ± 0.13 μg*/*mL	[100]
Ferrous ion chelating ability assay	*P. cembra* L.	Needle 80% aqueous methanol extract	EC50 = 1.755 ± 22 μg/mL	[104]
	*P. sylvestris* L.	Shoot aqueous extract	42.76 ± 5.7 μM FeSO_4_/g dw	[7]
	*P. sylvestris* L.	Air-dried shoot 40% aqueous ethanol extract	37.79 ±3.64 μM FeSO_4_/g dw	[105]
	*P. sylvestris* L.	Vacuum-dried shoot 40% aqueous ethanol extract	47.25 ±14.06 μM FeSO_4_/g dw	[105]
	*P. sylvestris* L.	Freeze-dried shoot 40% aqueous ethanol extract	21.79 ± 4.36 μM FeSO_4_/g dw	[105]
Superoxide anion	*P. radiata*	Aqueous bark extract	47.6 ± 5.8% at 23.0 ug/mL	[57]
	*P. brutia*	Bark 80%aqueous methanol extract	39.37 ± 0.85 μg/mL	[100]
Hydrogen peroxide	*P. radiata*	Aqueous bark extract	47.8 ± 12.3% at 8.0 ug/mL a	[57]
15-LO inhibition assay	*P. brutia*	Bark 80%aqueous methanol extract	EC50 = 22.47 ± 0.75 μg/mL	[100]

**Table 8 plants-10-01306-t008:** Commercial pharmacological products from pine.

Name of Formulation	Plant Part Used	Pharmacological Activity Declared by the Manufacturer	References
Polyherbal oil extract	Oleoresin of *P. roxburghii*	Analgesic and anti-inflammatory	[129]
Rumalaya gel	Resin from *P. roxburghii*	Lowers the joint and bone pain associated with various orthopedic ailments	[119]
Pycnogenol^®^	*P. pinaster bark*	Antimicrobial activity and treatment of asthma, Attention Deficit Hyperactivity Disorder, chronic venous insufficiency, diabetes, erectile disorders and osteoarthritis	[108]
Oligopin^®^	*P. pinaster bark*	Cardiovascular and vein health, antioxidant, treatment of male sexual disorders and ADHD (Attention Deficit Hyperactivity Disorder)	[130]
PineXol^®^	*P. densiflora* bark	Anti-inflammatory agent, enhances blood circulation and improves skin conditions	[146]
Flavangenol^®^	*P. maritima* bark	Lowers blood pressure and improves glycemic control, plasma lipoprotein profile, body weight, antioxidative capacity, level of anti-inflammatory markers and liver function tests	[147]
Enzogenol^®^	*P. radiata* bark	Antioxidant, anti-inflammatory, neuroprotective and anti-diabetic properties.	[148]

**Table 9 plants-10-01306-t009:** Application of *Pinus* in food products.

Food Application	Material Used	Application Result	References
Fruit juices supplementation	*P. pinaster* Ait bark extract	Fresh fruit juices enriched with PBE exhibited the highest inhibitory effect on the growth of pathogenic intestinal bacteria, primarily *E. coli* and *Enterococcus faecalis*. The in vitro digestion process reduced the antibacterial effect of juices on the majority of pathogenic bacteria by approx. 10%.	[152]
		ROS production increased in the inflamed cells exposed to digested commercial red fruit juice (86.8 ± 1.3%) in comparison with the fresh juice (77.4 ± 0.8%) and increased in the inflamed cells exposed to digested enriched red fruit juice (82.6 ± 1.6%) in comparison with the fresh enriched juice (55.8 ± 6%)	[158]
		Following the in vitro digestion, the level of detectable phenolic compounds (expressed as gallic acid equivalent) was higher in both pineapple and red fruit juices enriched with Pycnogenol than non-enriched commercial juices (155.6 mg/100 mL vs 94.6 mg/100 mL and 478.5 mg/100 mL vs 406.9 mg/100 mL respectively). Increased antioxidant activity (measured by 2,2’-azino-bis (3-ethylbenzothiazoline-6-sulphonic acid) (ABTS) and oxygen radical absorbance capacity (ORAC) methods) was observed in digested enriched juices, contrary to the same samples before digestion. Undigested, enriched with Pycnogenol pineapple juice displayed a higher antiproliferative effect between the 24th and 72nd hour of incubation in comparison with the non-enriched juice.	[153]
	*P. brutia*, *P. pinea* bark extracts, Pycnogenol^®^.	The paper shows that juices enriched with pine bark extracts exhibit higher antioxidant capacities and ascorbic acid contents compared to the control group, thereby providing improved functionality.	[154]
Yoghurt supplementation	French marine bark extract	Addition of Pycnogenol neither significantly affected the growth of microorganisms nor caused any modifications in nutritional parameters during the storage of yoghurt. Data indicate that neither the content of total polyphenol nor selected phenolic substances (catechin, epicatechins, chlorogenic acid and caffeic acid) was affected during the shelf life. In conclusion, these results indicate Pycnogenol as a valuable ingredient for the enrichment of yoghurt preparations.	[159]
	*P. nigra* cones	This study used yoghurt samples to identify the LAB strains generated by the pine cone addition and determined the physicochemical properties of these samples. The genotypic identification revealed that in yoghurt samples, *Streptococcus* thermophilus strains were the main force conducting the fermentation process, while Lactobacillus plantarum strains appeared in three yoghurt samples as an adjunct culture. The time of pine cones collection significantly affected the physicochemical properties of yoghurt.	[160]
Kefir	Pine bud syrup	The pine bud syrup used to enrich kefir contains a lot of polyphenols and terpenes, as well as exhibiting a high antioxidant activity. The addition of pine bud syrup resulted in an increase in total solids, as well as a decrease in the content of fat, proteins and pH levels. The kefir sample containing 10% pine bud syrup was the most appreciated by the sensory panel. Its overall acceptability score was higher (6.71 points) than that of the regular kefir (5.57 points). The addition of 10% pine bud syrup improved the texture and consistency of regular kefir.	[151]
Meat	Pine bark extract (Pycnogenol)	The pine bark extract (Pycnogenol^®^) significantly improved the oxidative stability of cooked beef and reduced the hexanal content by 73% after 3 days of refrigerated storage.	[155]
Tea	Pine needles	Supplementation of pine needle extract at 1, 2, 4 and 8% in the control diet and mixed groups significantly decreased the weight gain and visceral fat mass in comparison with the corresponding values of the control group.	[34]
Beer	*P. sylvestris* needles	The addition of needles increases the beer gustatory properties and decreases the methanol content. The content of ascorbic acid in ready-made drinks amounts to 3.52 mg/100 g. The antioxidant activity of elaborated beer is 178.1 C/100 g and determines its high biological value. In the study, the influence of beer enriched with needle extract was evaluated concerning the antioxidant system of organisms of biological objects. Under acute pathological conditions, a beer with needle extract decreases its oxidative influence on brains of the biological objects.	[161]
Bread	Fermented pine needle extract syrup	Bread with a higher content of pine needle extract syrup demonstrated a slower increase of bread hardening during the storage period, suggesting a slowdown of bread retrogradation. The addition of pine needle extract syrup in bread dough also inhibited the growth of aerobic bacteria and moulds on the bread surface (by 0.8~24 in log (CFU/g) during the 4-day storage). The use of concentration higher than 11% initially gave the bread a strong, fine needle flavour, which disappeared after 2 days. Generally, the addition of pine needle extract syrup had no negative effect on the quality (including sensory) of bread. Therefore, the addition of needle extract syrup could improve storage stability and extend the shelf life of bread.	[156]

Abbreviations: PBE—pine bark extract; ABTS—2,2′-azino-bis (3-ethylbenzothiazoline-6-sulphonic acid; ORAC—oxygen radical absorbance capacity; LAB—lactic acid bacteria; CFU—colony-forming unit.

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
