# Peer review of "Pinus Species as Prospective Reserves of Bioactive Compounds with Potential Use in Functional Food—Current State of Knowledge"

_plants, 2021, doi:10.3390/plants10071306_

Round 1

Reviewer 1 Report

Correct

There are two to eight needles per fascicle and the position of the resin canals is polymorphic (septa; internal, medial external c.); the seed wings are articulated or oppressed.

Line 61 - The collected data concern four species: Pinus contorta L. Pinus pinea L. Pinus sylvestris L. There are only three?.

Line 64 Delete - Different parts of the pine tree have different nutritional contents. The seeds have the 64 highest energy value due to high-fat content [Nergiz, and Dönmez, 2004]. The seeds also 65

Table 4 – Correct.

 2,5-dihydroxobenzoic acid - 2,5-dihydroxybenzoic acid

()-eipgallocatechin gallate - epigallocatechin gallate

ferrulic acid - ferulic acid

rosmaniric acid - rosmarinic acid

 narginin - naringin

monomethyl pinosylvin – correct - Pinosylvin 3-methyl ether or Pinosylvin monomethyl ether

correct dihydromonomethyl pinosylvin

Line 230 - Experiments revealed that bark extracts containing 20% of ethanol, 40% of ethanol and 20% of acetronil correct acetonitrile displayed the highest antioxidant potential and the highest content of phenolic compounds.

There are problems with Table 4 - Polyphenol content,  Table 5 - The examplary composition of essential oils extracted from pine. Table 7 -  Pharmacological properties of Pinus, Table 8. Commercial pharmacological products from pine and Table 9. Application of Pinus in food products. In my opinion, these tables need to be reorganized. It's confusing and hard to read.

Petroleum ether, ethyl acetate, chloroform and P. roxburghii Sarg. ethanolic extract - Petroleum ether and chloroform extracts were the only compounds (extracts) that exhibited anticancer activity.

The antimicrobial activity of extracts, oils and resins from trees of the Pinus genus may be related to various organic compounds, such as alkaloids (reference), phenols and terpenes (Table 4 and Table 5).

Piperidine alkaloids -  Phytochemistry Letters - Volume 26, August 2018, Pages 106-109

Line 308 - Compounds extracted from the trees of the Pinus genus presented in the study exhibited different levels of antimicrobial activity against yeast, gram-positive and gram-negative bacteria, which validates the traditional application of these substances. Additionally, such extracts, oils and resins display the insecticidal, phytotoxic and antioxidant potential. Therefore, it is necessary to conduct research aided by biological studies with recovery, identification and testing of a single compound and/or multiple compounds to determine its/their biological effects (Add references)

The percentage inhibition of PREO activity (essential oil) was found to be concentration-dependent (Figure 1 is a supplementary data? I did not find ).

Author Response

Thank you for your thorough review. All comments and recommendations indicated have been included in the revised version of the manuscript. Changes to the manuscript were made in the content of the publication. The comments listed in the reviews are referenced below with the sing ">>"

We hope that the current state of the manuscript meets the standards of the Journal and that publication will be possible.

Best regards, Marcin Dziedzinski
---------------------------------------
Answers: 
There are two to eight needles per fascicle and the position of the resin canals is polymorphic (septa; internal, medial external c.); the seed wings are articulated or oppressed.

>>That statement has been corrected

Line 61 - The collected data concern four species: Pinus contorta L. Pinus pinea L. Pinus sylvestris L. There are only three?.

 >>That statement has been corrected

Line 64 Delete - Different parts of the pine tree have different nutritional contents. The seeds have the 64 highest energy value due to high-fat content [Nergiz, and Dönmez, 2004]. The seeds also 65

>>That statement has been deleted

Table 4 – Correct.

 2,5-dihydroxobenzoic acid - 2,5-dihydroxybenzoic acid

()-eipgallocatechin gallate - epigallocatechin gallate

ferrulic acid - ferulic acid

rosmaniric acid - rosmarinic acid

 narginin - naringin

monomethyl pinosylvin – correct - Pinosylvin 3-methyl ether or Pinosylvin monomethyl ether

correct dihydromonomethyl pinosylvin

  >>That names have been corrected

Line 230 - Experiments revealed that bark extracts containing 20% of ethanol, 40% of ethanol and 20% of acetronil correct acetonitrile displayed the highest antioxidant potential and the highest content of phenolic compounds.

  >>That has been corrected

There are problems with Table 4 - Polyphenol content,  Table 5 - The examplary composition of essential oils extracted from pine. Table 7 -  Pharmacological properties of Pinus, Table 8. Commercial pharmacological products from pine and Table 9. Application of Pinus in food products. In my opinion, these tables need to be reorganized. It's confusing and hard to read.

>>Additional lines have been added to make tables more readable

Table 7 - P. densiflora aqueous bark extract (Anticholinesterase activity) is a neuroprotective effect

. >>That has been corrected

Petroleum ether, ethyl acetate, chloroform and P. roxburghii Sarg. ethanolic extract - Petroleum ether and chloroform extracts were the only compounds (extracts) that exhibited anticancer activity.

  >>That has been corrected

The antimicrobial activity of extracts, oils and resins from trees of the Pinus genus may be related to various organic compounds, such as alkaloids (reference), phenols and terpenes (Table 4 and Table 5).

Piperidine alkaloids -  Phytochemistry Letters - Volume 26, August 2018, Pages 106-109

  >>References have been added.

Line 308 - Compounds extracted from the trees of the Pinus genus presented in the study exhibited different levels of antimicrobial activity against yeast, gram-positive and gram-negative bacteria, which validates the traditional application of these substances. Additionally, such extracts, oils and resins display the insecticidal, phytotoxic and antioxidant potential. Therefore, it is necessary to conduct research aided by biological studies with recovery, identification and testing of a single compound and/or multiple compounds to determine its/their biological effects (Add references)

  >>References have been added.

The percentage inhibition of PREO activity (essential oil) was found to be concentration-dependent (Figure 1 is a supplementary data? I did not find ).

>>Sorry, that was a mistake. Figure 1 is not included.

Reviewer 2 Report

The review presented by Dziedziński et al. summarizes the potential of Pinus species usage in food and nutraceutical industry as antioxidants in order to improve its nutritional and bioactive quality. It contains useful information showing generally the positive impact of Pinus extract and essential oils addition on the functional properties of various products for further industrial applications. Overall, quality of the paper is very high. It is well written, carefully organized therefore I suggest some minor modifications which could improve this paper.

Table 1 – put “Pinus” in italic

Line 65 – correct the citation in the text “Nergiz, and Dönmez, 2004”. Moreover, part of this sentence is missing - “The seeds also”.

Table 4 – please explain the abbreviation “dw” in footnote of the table

Line 126 – delete “both”

Line 251 - in-vitro – in italic

Table 9 – all abbreviations used in Table 7 should be explained in footnote of the table

The authors listed all the benefits and advantages of the use of Pinus L. in the food and nutraceuticals industry, but it is necessary to add a paragraph that would include the cytotoxicity of this species.

Author Response

Thank you for your thorough review. All comments and recommendations indicated have been included in the revised version of the manuscript. Changes to the manuscript were made in the content of the publication. The comments listed in the reviews are referenced below with the sing ">>"

We hope that the current state of the manuscript meets the standards of the Journal and that publication will be possible.

Best regards, Marcin Dziedzinski
---------------------------------------

Table 1 – put “Pinus” in italic

>>That has been corrected

Line 65 – correct the citation in the text “Nergiz, and Dönmez, 2004”. Moreover, part of this sentence is missing - “The seeds also”.

>>That has been corrected

Table 4 – please explain the abbreviation “dw” in footnote of the table

>>Abbreviation explanation has been added.

Line 126 – delete “both”

>>That has been corrected

Line 251 - in-vitro – in italic

>>That has been corrected

Table 9 – all abbreviations used in Table 7 should be explained in footnote of the table

>>Abbreviations explanations have been added.

The authors listed all the benefits and advantages of the use of Pinus L. in the food and nutraceuticals industry, but it is necessary to add a paragraph that would include the cytotoxicity of this species.

>>The several cytotoxicity results are in the table 7 (against cancer cells), and general toxicity is mentioned in paragraph 210 „ [87]. Additional statement about GRAS status has been included.

This manuscript is a resubmission of an earlier submission. The following is a list of the peer review reports and author responses from that submission.